# Influence of the Thermal Environment on Occupational Health and Safety in Automotive Industry: A Case Study

**DOI:** 10.3390/ijerph19148572

**Published:** 2022-07-14

**Authors:** Constanța Rînjea, Oana Roxana Chivu, Doru-Costin Darabont, Anamaria Ioana Feier, Claudia Borda, Marilena Gheorghe, Dan Florin Nitoi

**Affiliations:** 1Faculty of Industrial Engineering and Robotics, University Politehnica of Bucharest, Splaiul Independenței Nr. 313, 060042 Bucharest, Romania; rinjea.tatiana@gmail.com (C.R.); virlan_oana@yahoo.co.uk (O.R.C.); ctistere@yahoo.com (C.B.); ghe.marilena@gmail.com (M.G.); nitoidan@yahoo.com (D.F.N.); 2National Research and Development Institute of Occupational Safety[—]INCDPM “Alexandru Darabont” Bucharest, 35A Ghencea Blvd., 061692 Bucharest, Romania; 3Materials and Manufacturing Engineering Department, Politehnica University Timișoara, Bd. Mihai Viteazu Nr. 1, 300222 Timișoara, Romania; anamaria.feier@upt.ro

**Keywords:** occupational health and safety, ergonomics, thermal environment, human thermal comfort, automotive industry

## Abstract

Considering thermal environment aspects have a major impact not only on occupational health and safety (OH&S) performance but also on the productivity and satisfaction of the workers, the aim of the case study was to assess the thermal comfort of a group of 33 workers in an automotive industry company, starting with collecting data about the thermal environment from different workplaces, continuing with the analytical determination and interpretation of thermal comfort using the calculation of the Predicted Mean Vote (*PMV*) and Predicted Percentage of Dissatisfied (*PPD*) indices, according to provisions of the standard ISO 7730:2005, and comparing the results with the subjective perception of the workers revealed by applying individual questionnaires. The results of the study represent an important input element for establishing the preventive and protective measures for the analysed workplaces in correlation with the measures addressing other specific risks and, also, could serve as a model for extending and applying to other similar workplaces in future studies. Moreover, the mathematical model and the software instrument used for this study case could be used in further similar studies on larger groups of workers and in any industrial domain.

## 1. Introduction

The thermal environment has a major impact on occupational health and safety (OH&S) and also has a great influence on the productivity and satisfaction of workers [1,2]. Occupational thermal stress, determined by an improper thermal environment, could lead to a higher risk of diseases and health problems, reduces employees’ work capacity and lowers productivity output [3]. Thermal stress has two components: Heat stress and cold stress, either of which occurs when the core body temperature is no longer maintained at 36–37 °C. Depending on the work environment, the body reacts to maintain its temperature so that the employee can perform optimally. A body adapts to hot temperatures by sweating and increasing skin blood flow to prevent the body temperature from rising, while in cold temperatures, the body shivers and reduces skin blood flow to prevent the body temperature from decreasing [4]. In extreme conditions, the adaptation and maintenance processes are not enough and start to fail, causing the body to undergo thermal stress, defined as the sum of the environmental and metabolic heat load imposed on the individual [4,5].

In recent years, the number of studies on the effects of the indoor thermal environment on human physical and mental health has increased. The studies focused on different aspects of the theme, such as assessing the indoor thermal environment in specific locations (classrooms, workplaces) [6,7], using electronic sensors for monitoring and improving the thermal sensation [8], using IT instruments for the modelling and prediction of thermal comfort [9,10,11], or the effects of heat-related illness on mental health [12].

ISO 7730:2005 Standard defines thermal comfort, also known as hygrothermal comfort, as “that condition of mind which expresses satisfaction with the thermal environment” [13], which is a subjective concept and thus very difficult to assess [2,10]. Thermal comfort is determined by six environmental and personal factors, which may be independent of each other but determine the employee’s thermal comfort by their synergic effects [14,15,16]. Thus, environmental factors include air temperature, radiant temperature, air velocity and humidity, while personal factors include clothing insulation and metabolic heat. Air temperature is the temperature of the air surrounding the body, given in degrees Celsius (°C). Radiant temperature is determined by the heat that radiates from a warm object and has a greater influence than the air temperature on how we lose or gain heat to the environment. Air velocity is the speed of air moving across the employee. Relative humidity is the ratio between the actual amount of water vapour in the air and the maximum amount of water vapour that the air can hold at that air temperature. Clothing insulation is an important determining factor for thermal comfort, as well as the ability of an employee to make reasonable adaptations to their clothing to feel comfortable and the thermal insulation properties of the personal protective equipment (PPE). The metabolic heat is directly related to the physical effort required by the work task [16]. Furthermore, in certain situations and especially in outdoor workplaces, the workplace thermal environment is influenced by climate change and the increasing global temperature [17].

The main effect of cold stress is hypothermia, a gradual process that is shown in three stages: Mild, moderate and severe. Mild-stage symptoms are shivering, grogginess and poor judgment or confused thinking. The moderate stage is manifested by violent shivering, inability to think or pay attention, slow, shallow breathing, slurred speech and poor body coordination. In the severe stage, the symptoms are loss of consciousness, little or no breathing and a weak, irregular or non-existing pulse [18].

Typical symptoms of heat stress are an inability to concentrate, muscle cramps, heat rash, severe thirst, fainting, heat exhaustion (fatigue, giddiness, nausea, headache, moist skin) and heat stroke (hot dry skin, confusion, convulsions and eventual loss of consciousness) [19,20].

Physiological effects of thermal stress exposure determine a higher risk of work accidents, diseases and health problems, affecting productivity by presenteeism and absenteeism issues and generating additional costs for the companies [21,22]. Furthermore, thermal stress exposure could have a significant negative synergic effect with other types of occupational risks such as neuropsychic effort in the case of workplaces where the mental load required by the work tasks is at a high level (e.g., air traffic controllers or electricians working on powerlines) [23,24,25].

Controlling thermal comfort can be realized by both technical and organizational measures, such as controlling the environment by adjusting its parameters (temperature, humidity and air velocity) as appropriate; separating the source of heat or cold from the employee; controlling the task (e.g., restrict the length of time that employees are exposed to hot or cold conditions and introduce mechanical aids); controlling clothing (e.g., by providing appropriate PPE and allowing employees to adapt their clothing where possible); allowing the employee to make behavioural adaptations (e.g., use personal heaters or fans, allow employees to adjust thermostats or open windows as appropriate); monitoring the employee (by providing appropriate supervision and training as well as obtaining medical advice from an occupational health professional for employees in sensitive risk groups, for instance, workers with disabilities, young and old workers, pregnant and nursing mothers, etc.) [26,27].

The aim of the work is to assess the thermal comfort of a group of workers in an automotive industry company, starting with collecting data about the thermal environment from different workplaces, continuing with the analytical determination and interpretation of thermal comfort using the calculation of the Predicted Mean Vote (*PMV*) and Predicted Percentage of Dissatisfied (*PPD*) indices, according to provisions of the standard ISO 7730:2005, and comparing the results with the subjective perception of the workers revealed by applying individual questionnaires. The need for this study was raised by the finding that in the OHS risk assessment for the analysed workplaces, the risk related to the thermal environment was assessed at a low level, but some workers complained about thermal comfort, and there were no available measurements of the thermal environment parameters. The results of the study represent an important input element for establishing the preventive and protective measures for the analysed workplaces and could also serve as a model for extending and applying them to other similar workplaces in future studies.

## 2. Materials and Methods

### 2.1. Mathematical Model

Thermal comfort is assessed by determining the *PMV* using a thermal sensation scale divided into 7 classes, as shown in Figure 1: +3 (Hot), +2 (Warm), +1 (Slightly warm), 0 (Neutral), −1 (Slightly cool), −2 (Cool), −3 (Cold) [2,13].

*PMV* predicts the mean value of the votes of a large group of persons, and its value depends on six parameters: Air temperature, radiant temperature, air velocity, relative humidity, activity level and clothing insulation, as shown in Fanger’s equations [2,13]:(1)$$PMV=\left(0.303e^{-0.036M}+0.028\right)\{\left(M-W\right)-3.05\cdot 10^{-3}\left[5733-6.99\left(M-W\right)-p_{a}\right]-0.42\left[\left(M-W\right)-58.15\right]\;-1.7\cdot 10^{-5}M\left(5867-p_{a}\right)-0.0014M\left(34-t_{a}\right)-3.96\cdot 10^{-8}f_{cl}\left[{\left(t_{cl}+273\right)}^{4}-{\left(t_{mr}+273\right)}^{4}\right]-f_{cl}h_{c}\left(t_{cl}-t_{a}\right)\}$$
(2)$$t_{cl}=35.7-0.028\left(M-W\right)-I_{cl}\left\{3.96\cdot 10^{-8}f_{cl}\left[{\left(t_{cl}+273\right)}^{4}-{\left(t_{mr}+273\right)}^{4}\right]+f_{cl}h_{c}\left(t_{cl}-t_{a}\right)\right\}$$
(3)$$h_{c}=\{\begin{array}{cc} 2.38{\left|t_{cl}-t_{a}\right|}^{0.25}, & for\;2.38{\left|t_{cl}-t_{a}\right|}^{0.25}>12.1\sqrt{v_{ar}}\\ 12.1\sqrt{v_{ar}}, & for\;2.38{\left|t_{cl}-t_{a}\right|}^{0.25}<12.1\sqrt{v_{ar}} \end{array}$$
(4)fcl={1.00+1.290Icl,for Icl≤0.078 m2·KW1.05+0.645Icl,for Icl>0.078 m2·KW
where:*M* is the metabolic rate, in W/m^2^;*W* is the effective mechanical power, or external work, in W/m^2^;*I_cl_* is the clothing insulation, in m^2^·K/W;*f_cl_* is the clothing surface area factor;*t_a_* is the air temperature, in °C;*t_mr_* is the mean radiant temperature, in °C;*v_ar_* is the relative air velocity, in m/s;*p_a_* is the water vapour partial pressure, in Pa;*h_c_* is the convective heat transfer coefficient, in W/(m^2^ K);*t_cl_* is the clothing surface temperature, in °C.

The value of *M* is established by Annex B of ISO 7730:2005, depending on the activity type. The value of *W* is equal to zero for most activities [2,13]. The value of *I_cl_* is provided by Annex C of ISO 7730:2005. The values of *t_a_*, *v_ar_* and relative humidity (*RH*, related to *p_a_*) are determined by measurements at the analysed workplace. The value of *t_mr_* was determined with a heat stress monitor by measuring the *t_a,_ v_ar_* and the global temperature, and using the equation provided by ISO 7726:1998, Annex B [28]. The value of *t_cl_* and *h_c_* could be determined by iterations from Equations (2) and (3).

The *PPD* index establishes a quantitative prediction of the percentage of thermally dissatisfied people who feel too cool or too warm, and is determined using the following equation [13]:(5)$$PPD=100-95\cdot e^{\left(-0.03353\cdot PMV^{4}-0.2179\cdot PMV^{2}\right)}$$

### 2.2. Software Instrument

A software instrument was elaborated by the authors, using Microsoft^®^ Excel^®^, for the calculation of the *PMV* and *PPD* values for an assessed workplace based on the mathematical model presented above. The caption of the sheet is presented in Figure 2, and the calculated cells and their corresponding formulas are presented in Table 1.

### 2.3. Checklist for Estimating the Level of Thermal Comfort

The checklist adapted from the Health and Safety Executive, shown in Table 2, was used to estimate the individual perception of workers about the workplace’s thermal comfort at the moment when the measurements were performed [29].

### 2.4. Analysed Workplaces

The case study was performed in a Romanian automotive industry company by analysing 33 workbenches from different production lines from the perspective of thermal comfort. The workers of this company were selected as the target participants of the study considering that the company was a partner in larger research on OHS, which includes the present study. Even if there is no registration of work accidents or occupational diseases caused directly by thermal stress, some of the workers raised an issue with thermal comfort. The analysed workplaces are located on the same open-space production floor. The target group is representative of the production floor, as it represents all of the workers at this production site. The measurements were performed simultaneously with 3 sets of instruments on one experimental day, in May, between 11 a.m. and 1 p.m., in sunny weather conditions with an outdoor temperature of 22 °C. The production floor is equipped with an air conditioning system, but the system was off at the moment of measurements. The thermal environment conditions were measured for each workplace using a heat stress monitor and an air velocity measuring instrument. The measurement height was 1.1 m since the workers’ activity is performed in an orthostatic position. The workplaces and the relevant information about workers (age and gender) are presented in Table 3. The scheme of the workplaces’ location on the production floor is represented in Figure 3. 

The activity performed by the employees at workbenches consists of manual handling and manual assembly of small dimensions components, using a scanner, putting the final products in boxes and using a trolley to transport boxes to the warehouse.

The PPEs worn by the workers consist of safety glasses, hearing protectors, protective gloves and safety shoes.

The work clothes worn by the workers consist of a cotton T-shirt, trousers, a jacket, underwear, socks and protective shoes and are similar for all the workers.

For each workplace, the thermal environment parameters (*t_a_*, *t_mr_*, *v_ar_* and *RH*) were determined by measurements made with a heat stress monitor and an air velocity measuring instrument. The heat stress monitor was equipped with the following sensors: A globe thermometer (accuracy ±0.5 °C between 0 °C and 120 °C), relative humidity sensor (accuracy ±0.5% between 20% and 90%) and dry bulb thermometer (accuracy ±0.5 °C between 0 °C and 120 °C). The air velocity sensor accuracy was ±0.05 m/s, and the range was 0 to 20 m/s.

Then, based on the description of the activity presented above, the PPEs and work clothes, and using the provisions of the standard ISO 7730:2005, the values for the following parameters were established and used for all assessed workplaces, considering their similarity:(6)$$M=116\;\left[\frac{W}{m^{2}}\right]$$
(7)W=0
(8)$$I_{cl}=0.155\;\frac{m^{2}K}{W}=1\;\mathrm{clo}$$

In the next step, using the presented software instrument, the values of *PMV* and *PPD* were calculated, and the thermal sensation was determined for each workplace based on the *PMV* value. Furthermore, each employee from the analysed workplaces was invited to complete the questionnaire presented in Table 2, indicating the relevant information: Workplace identification (line and workbench numbers), age and gender. The worker’s perception of the risk related to thermal comfort issues was expressed by the number (*N*) of questions answered with “Yes”. The higher the value of *N*, the higher the level of risk perceived by the worker related to thermal comfort issues.

## 3. Results

### 3.1. Thermal Comfort Assessment 

The measurement results and collected data are presented in Table 4.

### 3.2. Interpretation of the Results 

For the assessed workplaces, the value of *PMV* ranges from 1.23 (for WP27) to 1.73 (for WP11) with corresponding *PPD* values of 31.28% and 48.83%, respectively. The thermal sensation is determined as ”Slightly warm” for 21 workplaces and as ”Warm” for 12 workplaces. These results show a moderate risk of thermal stress for the workers, from the point of view of the thermal environment parameters.

However, based on their responses to the questionnaire, the workers perceive the work environment differently from the point of view of their thermal comfort. Thus, the analysed worker group consists of 33 employees, with 14 women and 19 men aged from 20 years (for WP1) to 62 years (for WP33). The workers’ responses were analysed in two separate groups: The first group is formed by the workplaces where the thermal sensation was determined as ”Slightly warm” and the second group is formed by the workplaces where the thermal sensation was determined as ”Warm”. For each group, the responses were clustered by age into three subgroups (20 to 35 years, 36 to 50 years and over 50 years), and then each of these subgroups was divided in series by gender. For each series, the average value of *N*, noted with *N_avg_*, was calculated. The results are shown in Table 5.

The data presented in Table 5 show a different personal perception of the workers from the point of view of thermal comfort, even if they perform their activity in workplaces with similar thermal sensation conditions, determined by the *PMV* value, which is calculated based on the measured parameters and provisions in the ISO 7730:2005. Thus, for each of the two groups differentiated by thermal sensation, *N_avg_* and the level of the risk related to thermal comfort issues are lower for the young worker group (20–35 years) and higher for the middle-aged (36–50 years) and elderly workers (over 50 years). Furthermore, inside the same age subgroup, the perception of men and women is similar in most cases.

For the workplaces analysed in this study case, the following conclusions could be synthetised based on the obtained results:The thermal sensation levels, determined by *PMV* values, are too high to ensure thermal comfort for most workers and show a moderate risk of thermal stress for the workers, in all analysed workplaces.The middle-aged and elderly workers in comparison with younger workers are more susceptible to being the subject of work accidents, near-misses and occupational diseases indirectly generated by thermal stress, considering they perceive poorer thermal comfort than younger workers, as the responses to the questionnaire indicate. This perception could be the cause of errors due to premature fatigue; this is the situation for workplaces WP 02, WP 03, WP 09, WP 10, WP 11, WP 14, WP 15, WP 16, WP 20, WP 21, WP 22, WP 23, WP 25, WP 26, WP 28, WP 29, WP 30, WP 31, WP 32 and WP 33.Even if the premature fatigue due to poor thermal comfort perceived by the middle-aged and elderly workers does not generate OH&S issues, it may still be the main cause of worker errors, which could generate productivity or product quality issues; thus, the thermal comfort at the workplace not only represents an OH&S issue but equally a productivity and quality issue.Both *PMV* values and workers’ responses to the questionnaire indicate the necessity to establish and implement a programme of technical and organizational preventive and protective measures to improve the thermal comfort conditions in all analysed workplaces, but especially in the workplaces with middle-aged and elderly workers.The findings on the thermal comfort of the workers in the analysed workplaces should be integrated into the overall OH&S risk assessment for these workplaces; the potential synergic effect of thermal comfort issues with other risks, such as neuropsychic effort determined by the work tasks or medical conditions of the workers, should be considered.

## 4. Discussion

### 4.1. Thermal Environment Influence on OHS 

In any work system, four specific components are involved: Executor, workload, means of work (work equipment) and work environment of the workplace/workstation. These components are in a permanent state of interdependence [30,31,32,33]. Each of these work system components may generate dysfunction, leading to accidents and occupational diseases [34]. These dysfunctions can also be generated when workload, work equipment or work environment are not ergonomically adapted to the executor (worker). If the thermal environment, as a part of the work environment, is not adapted to the executor, he or she could undergo thermal stress, which can determine physiological effects that increase the probability of wrong actions or oversight of the worker. These worker errors can affect both a company’s OH&S performance (by producing incidents, work accidents or occupational diseases) and productivity (by generating products with quality issues, reducing worker’s performance or generating presenteeism and absenteeism issues) and also generate additional costs for companies [21,22]. Moreover, the potential negative synergic effects of thermal stress exposure and other types of occupational risks should be considered. Thus, a work environment temperature that is too low or too high reduces the worker’s resilience to neuropsychic effort in workplaces requiring mental load at a high level or could affect human comfort perception in the context of whole-body vibration [23,24,25,35], and the combined effects of hand-arm vibration and low temperature might lead to occupational hazards such as vibration-induced white finger syndrome (Raynaud’s syndrome) in workers [36,37]. Moreover, the impact of the thermal environment may, for example, vary according to a number of individual characteristics of the subjects, such as age, gender, fitness, medical conditions, etc. [38,39]. Understanding individual risks due to thermal stress is complicated by specific combinations of age- and gender-related differences in heat response and differences in heat exposure patterns. For many decades, mathematical models of the human thermal response have been used to simulate human physiological and thermoregulatory responses under different environmental conditions and activity levels. Problematic for models is the variation of thermoregulatory responses between different individuals, even under the same environmental conditions. In recent years, information on age, gender and body-mass-based differentials in heat responses has emerged [40]. Important individual characteristics influencing the thermal comfort perception are represented by the body surface area, metabolic rate and sweat rate, and all these three parameters are related to age and gender [40,41]. The results of this study case confirm the correlation between thermal comfort perception on one hand, and age and gender on the other hand.

### 4.2. Practical Implication 

For the analysed workplaces, the results of the study show a moderate risk of thermal stress, reflected by both thermal sensation levels determined by *PMV* values, and subjective perception of the workers determined by questionnaires. This situation could have a negative impact on the OHS level of the analysed workplaces. Thus, from an occupational health perspective, the main effect could be premature fatigue due to the poor thermal comfort perceived, especially by middle-aged and elderly workers. Furthermore, thermal stress issues could have a potential synergic effect on other risks, such as neuropsychic effort determined by the work tasks or medical conditions of the worker. From the occupational safety perspective, even if, for analysed workplaces, the thermal stress is at a moderate level and is not susceptible to directly generating work accidents, it could still be an indirect cause of a work accident by increasing the occurrence of worker errors generated by premature fatigue. Moreover, by generating premature fatigue, the thermal stress could be the cause of productivity or product quality issues. Thus, the thermal environment represents an important part of the work environment, as a component of the work system, and plays a significant role in the workplace’s OHS level by influencing the executor’s behaviour within the work system. Therefore, the OHS risk assessment should also be based on the measurement of the thermal environment parameters. After performing the OHS risk assessment, a preventive and protective plan should be elaborated on, containing organizational, technical, sanitary and other measures to improve the workplace OHS level.

One of the limitations of the study is the small size of the target group, which, even if it is representative of the analysed workplaces and suitable for establishing OHS preventive and protective measures, is not big enough for extrapolation. Another limitation is determined by performing the experiment in one day, considering that the thermal comfort perceived by workers depends on psychological factors and may also be different if the experiment had been performed across different days.

Monitoring the thermal environment can be performed by periodic measurements of thermal-comfort-determining parameters: Air temperature, radiant temperature, air velocity and humidity, and analysing the results in correlation with workers’ personal factors, such as clothing insulation and metabolic heat. Considering the necessary time-lapse to perform the measurements and process the data, this approach does not allow real-time monitoring. This type of monitoring is suitable for applications such as this study case, where the above-mentioned parameters are relatively constant during the work shift and may vary with season. When the environmental and personal factors register important variations during the work shift, real-time monitoring may be necessary, which can be performed using wearable sensors and devices, which can integrate different parameters from the environmental, behavioural and physiological domains, while also providing real-time monitoring of the worker’s health [42,43].

Based on the monitoring results, both technical and organizational measures should be established to improve the thermal comfort of workers. These measures should be integrated into the preventive and protective plan for the workplace, also containing measures that address other specific risks.

For the analysed workplaces, examples of technical measures are the following:Installing air conditioning and ventilation systems to ensure fresh air and suitable control of the temperature.Providing proper maintenance of the air conditioning and ventilation systems.Selecting and providing the PPEs by taking into account workers’ thermal comfort in addition to the requirements of protection against other specific risks.

For the analysed workplaces, examples of organizational measures are the following:Allowing the workers to adapt their clothing to the environment temperature.Allowing the workers, and stimulating them, to report any situation of thermal stress they undergo at the workplace.Providing periodic medical control of the workers and monitoring health issues that represent a risk factor when working in thermal stress.Providing proper training to workers on thermal stress and measures to avoid it.Considering reducing or shifting working time during heatwaves.Involving workers or their representatives in consultation and participation processes regarding OH&S issues.

## 5. Conclusions

The results of the study represent an important input element for establishing the preventive and protective measures for the analysed workplaces in correlation with the measures addressing other specific risks and could also serve as a model for extending and applying this to other similar workplaces in future studies. 

As the analysed worker groups are not big enough for extrapolation, the conclusion should be limited to this particular study case. However, the mathematical model and instruments (questionnaire and software) used for this study case could be used in further studies on larger groups of workers in any industrial domain.

## Figures and Tables

**Figure 1 ijerph-19-08572-f001:**
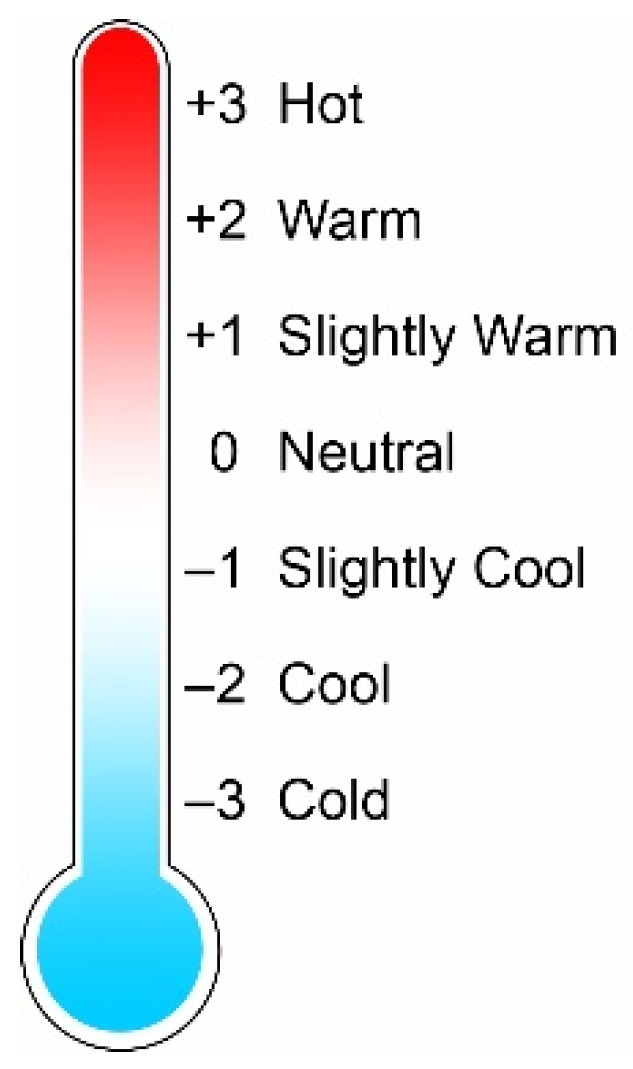
Seven-point thermal sensation scale.

**Figure 2 ijerph-19-08572-f002:**
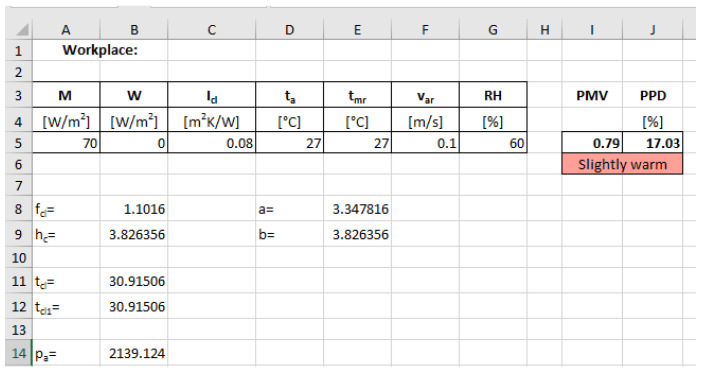
Software instrument for calculating the values of *PMV* and *PPD*.

**Figure 3 ijerph-19-08572-f003:**
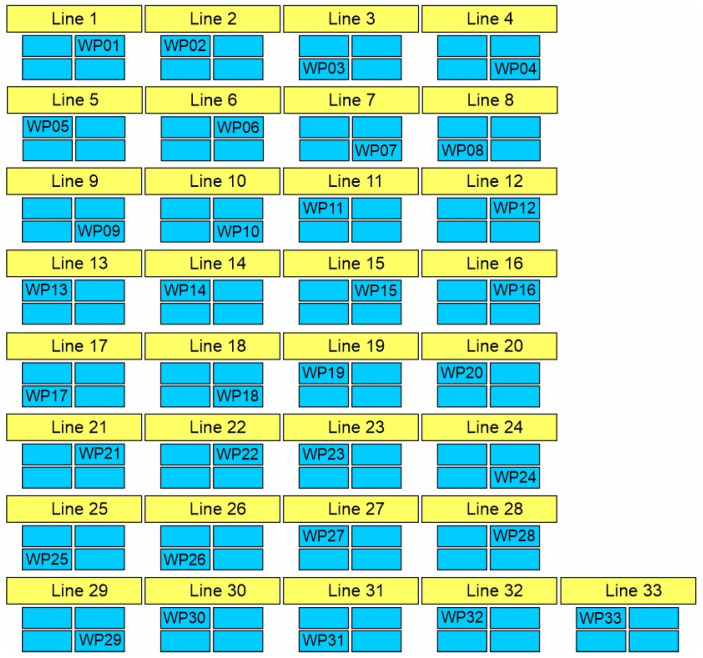
Scheme of the workplaces’ location on the production floor.

**Table 1 ijerph-19-08572-t001:** Calculated cells and their corresponding formulas.

Cell	Formula
B8	=IF(C5 <= 0.078;1 + 1.29*C5;1.05 + 0.645*C5)
B9	=IF(E8 > E9;E8;12.1*SQRT(F5))
E8	=2.38*ABS(B11 − D5)^0.25
E9	=12.1*SQRT(F5)
B12	=35.7 − 0.028*(A5 − B5) − C5*(3.96*10^(−8)*B8*((B11 + 273)^4 − (E5 + 273)^4) + B8*B9*(B11 − D5))
B14	=G5*10*EXP(16.6536 − 4030.183/(D5 + 235))
I5	=(0.303*EXP(−0.036*A5) + 0.028)*(A5 − B5 − 3.05*10^(−3)*(5733 − 6.99*(A5 − B5) − B14) − 0.42*(A5 − B5 − 58.15) − 1.7*10^(−5)*A5*(5867 − B14) − 0.0014*A5*(34 − D5) − 3.96*10^(−8)*B8*((B11 + 273)^4 − (E5 + 273)^4) − B8*B9*(B11 − D5))
I6	=IF(ROUND(I5;0) = 3;“Hot”;IF(ROUND(I5;0) = 2;“Warm”;IF(ROUND(I5;0) = 1;“Slightly warm”;IF(ROUND(I5;0) = 0;“Neutral”;IF(ROUND(I5;0) = −1;“Slightly cool”;IF(ROUND(I5;0) = −2;“Cool”;IF(ROUND(I5;0) = −3;“Cold”)))))))
J5	=100−95*EXP(0.003353*I5^4−0.2179*I5^2)

**Table 2 ijerph-19-08572-t002:** Thermal comfort checklist.

Factor	Description	Response (Yes/No)
Air temperature	Q1. Does the air feel warm or hot?	
	Q2. Does the temperature in the workplace fluctuate during a normal working day?	
	Q3. Does the temperature in the workplace change a lot during hot or cold seasonal variations?	
Radiant temperature	Q4. Is there a heat source in the environment?	
	Q5. Is there any equipment that produces steam?	
	Q6. Is the workplace affected by external weather conditions?	
Humidity	Q7. Are you wearing PPE that is vapour impermeable?	
	Q8. Do you complain that the air is too dry?	
	Q9. Do you complain that the air is too humid?	
Air movement	Q10. Is cold or warm air blowing directly into the workspace?	
	Q11. Are you complaining of draught?	
Metabolic rate	Q12. Is work rate moderate to intensive in warm or hot conditions?	
	Q13. Are you sedentary in cool or cold environments?	
PPE	Q14. Is PPE being worn that protects against harmful toxins, chemicals, asbestos, flames, extreme heat, etc.?	
	Q15. Is it difficult for you to make individual alterations to the clothing in response to the thermal environment?	
	Q16. Is respiratory protection being worn?	
What your employees think	Q17. Do you think that there is a thermal comfort problem?	

**Table 3 ijerph-19-08572-t003:** Analysed workplaces.

Age Subgroup	Gender	Number of Workers	Workplaces
20–35 years	M	6	WP04, WP07, WP12, WP17, WP19, WP24
	F	5	WP01, WP05, WP08, WP18, WP27
36–50 years	M	5	WP09, WP11, WP23, WP25, WP28
	F	5	WP06, WP10, WP13, WP26, WP29
Over 50 years	M	8	WP02, WP03, WP15, WP16, WP20, WP30, WP31, WP32
	F	4	WP14, WP21, WP22, WP33

**Table 4 ijerph-19-08572-t004:** Measurement results and collected data.

Code	*M* [W/m^2^]	*W* [W/m^2^]	*I_cl_* [m^2^K/W]	*t_a_* [°C]	*t_mr_* [°C]	*v_ar_* [m/s]	*RH* [%]	*PMV*	Thermal Sensation	*PPD*	Worker’s Age	Worker’s Gender	** *N* **
WP 01	116	0	0.155	24.9	25.0	0.10	44.8	1.38	Slightly warm	36.39%	20	F	3
WP 02	116	0	0.155	24.8	25.0	0.11	44.9	1.36	Slightly warm	35.80%	53	M	9
WP 03	116	0	0.155	24.9	25.0	0.08	44.7	1.40	Slightly warm	37.09%	60	M	9
WP 04	116	0	0.155	25.1	25.0	0.17	45.0	1.34	Slightly warm	35.23%	27	M	3
WP 05	116	0	0.155	25.2	25.0	0.13	45.1	1.38	Slightly warm	36.46%	35	F	2
WP 06	116	0	0.155	25.3	25.0	0.13	45.2	1.39	Slightly warm	36.78%	38	F	2
WP 07	116	0	0.155	27.0	27.0	0.07	44.7	1.69	Warm	47.69%	25	M	3
WP 08	116	0	0.155	26.7	27.0	0.21	44.5	1.59	Warm	44.00%	21	F	3
WP 09	116	0	0.155	26.1	26.0	0.24	44.2	1.46	Slightly warm	39.39%	43	M	8
WP 10	116	0	0.155	27.0	27.0	0.03	44.9	1.70	Warm	48.06%	48	F	8
WP 11	116	0	0.155	27.3	27.0	0.05	44.7	1.73	Warm	48.83%	47	M	9
WP 12	116	0	0.155	27.1	27.0	0.02	44.5	1.71	Warm	48.25%	33	M	2
WP 13	116	0	0.155	26.4	26.0	0.15	44.5	1.53	Warm	41.82%	37	F	2
WP 14	116	0	0.155	26.3	26.0	0.06	44.3	1.58	Warm	43.69%	51	F	8
WP 15	116	0	0.155	26.2	26.0	0.12	44.5	1.53	Warm	41.82%	54	M	9
WP 16	116	0	0.155	26.2	26.0	0.16	44.5	1.51	Warm	41.01%	58	M	9
WP 17	116	0	0.155	27.3	27.0	0.45	44.5	1.59	Warm	44.15%	26	M	3
WP 18	116	0	0.155	27.3	27.0	0.46	44.2	1.59	Warm	44.03%	22	F	3
WP 19	116	0	0.155	25.4	25.0	0.08	44.7	1.43	Slightly warm	38.40%	30	M	2
WP 20	116	0	0.155	25.5	25.0	0.19	44.6	1.37	Slightly warm	36.09%	60	M	9
WP 21	116	0	0.155	25.6	25.0	0.22	44.6	1.37	Slightly warm	35.96%	57	F	9
WP 22	116	0	0.155	25.5	25.0	0.04	44.8	1.46	Slightly warm	39.22%	59	F	8
WP 23	116	0	0.155	26.3	26.0	0.12	44.1	1.53	Warm	42.03%	45	M	9
WP 24	116	0	0.155	26.3	26.0	0.35	44.2	1.45	Slightly warm	38.95%	23	M	3
WP 25	116	0	0.155	26.3	26.0	0.25	43.8	1.47	Slightly warm	39.90%	44	M	8
WP 26	116	0	0.155	24.0	24.0	0.04	44.8	1.27	Slightly warm	32.69%	49	F	9
WP 27	116	0	0.155	24.4	24.0	0.15	44.6	1.23	Slightly warm	31.28%	34	F	2
WP 28	116	0	0.155	25.9	26.0	0.35	44.4	1.41	Slightly warm	37.43%	41	M	8
WP 29	116	0	0.155	25.6	25.0	0.02	44.5	1.46	Slightly warm	39.45%	40	F	8
WP 30	116	0	0.155	25.7	25.0	0.07	44.7	1.46	Slightly warm	39.53%	52	M	9
WP 31	116	0	0.155	25.3	25.0	0.06	44.4	1.44	Slightly warm	38.58%	61	M	8
WP 32	116	0	0.155	25.7	25.0	0.11	44.5	1.43	Slightly warm	38.32%	55	M	9
WP 33	116	0	0.155	25.3	25.0	0.09	44.4	1.41	Slightly warm	37.74%	62	F	9

**Table 5 ijerph-19-08572-t005:** Analysis of the workers’ responses to the questionnaire.

Thermal Sensation	Age Subgroup	Worker’s Gender	Number of Workers	*N_avg_*
Slightly warm	20–35 years	M	3	2.67
		F	3	2.33
	36–50 years	M	3	8.00
		F	3	6.33
	Over 50 years	M	6	8.83
		F	3	8.67
Warm	20–35 years	M	3	2.67
		F	2	3.00
	36–50 years	M	2	9.00
		F	2	5.00
	Over 50 years	M	2	9.00
		F	1	8.00

## Data Availability

Not applicable.

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
