# Peer review of "Influence of the Thermal Environment on Occupational Health and Safety in Automotive Industry: A Case Study"

_ijerph, 2022, doi:10.3390/ijerph19148572_

Round 1

Reviewer 1 Report

1.     The reasons for why selecting the workers of this company as the target participants were not stated. What’s the representativeness of this group of participants?

2.     There were only 33 participants in the case study. Whether the sample size was statistically significant. How to prove the significance of the sample size? 

3.     The differences in gender and age-subgroup were mentioned in the results; however, these findings seemed to be neglected in the discussion section. It is important to explain the underlying reasons for such findings.

4.     The focus of this study was the impact of the thermal environment on occupational health and safety. But, there was not much discussion on this relationship based on your findings. You provided a lot of information on the measurement results and result interpretation. However, your discussions seemed to be too brief. More elucidations on how thermal environments influence OSH regarding the findings can be discussed. 

5.     Sub-headings can be added in the discussion section. For example, the sub-heading of the last two paragraphs could be Practical Implication. 

Overall, this study is interesting, especially the way of conducting the experiment. The thermal environment is one of the vital elements that enormously affected the occupational health and safety of the workers. However, the relevant issues seemed seldom to be mentioned in previous related issues. This study can raise awareness of the importance of the thermal environment on the OSH. Nonetheless, some parts of the study were not comprehensive. Please read through your article again to enhance its quality. 

Reviewer 2 Report

1.       Title: consider changing the title because no data about occupational health and safety for example occupational diseases, accidents or injuries are collected and reported in this study. The focus of the study is on thermal comfort and discomfort.

2.       Introduction: line 31-92, thermal stress including heat and cold stress, and related fundamental factors are well reviewed. But why this study is needed and what have already been done, and what remains to be done are not described.

3.       Method: line 144-146, why the thermal sensation scale in figure 1 was not used to record workers’ thermal comfort and then directly compared with PMV calculations?

4.       Line 148-150, in which month of the year was this performed? Add weather conditions. Was the workplace air conditioned or heated? Were the thermal environment conditions measured? If so, how were the conditions measured? what instruments were used?

5.       Line 159, did all workers wear the same work clothing? What about other garment items such as underwear, socks, shoes, etc.?

6.       Table 3 should be summarized as group information. The details are not necessary.

7.       Results: line 162-163, how were the thermal environments parameters measured? which instruments were used? See comment above.

8.       Eq. 6-8 are redundant.

9.       Line 171-173, double check this. If Question #15 is answered yes, it means that the worker can adjust the clothing in response to the environment. Why does this adjustment increase thermal comfort risk? Q4: if there is a heat source in winter season, it will not increase thermal comfort risk. Q10: in hot environments, blowing cold air may help to improve thermal comfort, visa versa in cold environments, blowing warm air may also improve thermal comfort.

10.   Table 4: clarify if the thermal sensation is determined by PMV calculations or is perceived by the workers (rated by the workers using the thermal sensation scale in figure 1)?

11.   Line 197, change “if” to “in”

12.   Line 191-192, table 5, the number of workers is very small when the small sample of workers is sub-divided into 6 sub-groups. It is uncertain if the difference of the N_average is statistically significant.

13.   Conclusions: line 208- 213, this conclusion is speculation, not well supported by the data. Slightly warm or warm thermal sensation does not directly cause accidents or occupational diseases.

14.   Discussion: line 230-248, the discussion should focus more on thermal comfort and discomfort, explanation of the results and comparison with previous studies, and on highlighting new findings. The discussion here is more about extreme environments. For hot and cold environments, other thermal models and indices for example WBGT, PHS, IREQ should be used to evaluate heat stress or cold stress.

15.   Line 287-292, consider removing this paragraph because these are not conclusions.

Reviewer 3 Report

The present study investigates the thermal comfort conditions in an automotive company, combining measurements, for the calculation of the PMV and PPD indices, and the thermal comfort perception of the workers. Although this is a case study with probably just one day of experiment (it is not clearly stated) and the thermal perception of 33 workers, this topic is of great interest for many users. An in-depth analysis of the manuscript reveals some criticalities that should be addressed. Please find below specific comments and suggestions.

1)     Since the parameters of air temperature, air velocity, relative humidity and radiant temperature/globe temperature were measured, the used instrumentation should be described. This information is missing from the paper. Therefore, just describe the instruments and the type of the sensors and give measuring range and accuracy for the sensors. Please avoid mentioning any trade names in the paper. In addition, in the lines 298-300, it is stated that “the mathematical model and instruments used for this study case, could be used in further studies…”. In which instruments are you referring to? There is no any documentation about the used instruments in the manuscript.

2)     In the lines 128-130, it is stated that “The value of tmr is measured using a heat stress monitor or estimated by using a black-globe thermometer and an equation provided by the standard ISO 7726:1998 [22]”. This statement probably refers to what the ISO standard proposes. However, there is no documentation in the manuscript in which way the mean radiant temperature was calculated or measured. Did you directly measure the mean radiant temperature (if so, with which sensor) or did you measure the globe temperature, and the measured value was used for the calculation of mean radiant temperature using a formula? This information is crucial, since mean radiant temperature may, in some cases, be more important than air temperature.

3)     Important information concerning the experiments is missing. At what height were the measurements performed? Which month of the year were the experiments carried out? What time of the day were the experiments carried out? Were the measurements performed simultaneously during the experimental day? Were there any ventilation system or air-conditioning units installed in the space? Did you have multiple instrumentation installed in the space? Is this a one-day experiment? The experimental process/campaign along with the description of the examined space should be clearly presented in the methods and materials section.

4)     A photo showing the installed instrumentation in the workplace during the experiment could be useful and informative (I understand if you do not have one). A graph showing the layout of the space could also be used along with the measuring point/points and the workbenches. Were all the workbenches in the same wide space or in different rooms (production lines) or levels?

5)     Please avoid using different symbols for the decimal separator. For example, the decimal separator of the numbers presented in lines 177-178 and in Figure 2 is comma. However, the dot symbol is used in Table 4 and in all formulas. Please decide for one format and use consistently throughout the manuscript (text, tables, figures).

6)     A paragraph referring to other studies that investigate the thermal comfort conditions (based on measurements or/and subjective perception) in other workplaces or such types of indoor settings should be added in the introduction section.

7)     Table 4: The abbreviation of the parameters in the first line of Table 4 should be accompanied with the units, regardless the fact that their units were mentioned in the text.

8)     The text between the lines 162 and 173 does not describe or present results. Therefore, we would ask the authors to consider moving this part of the text to the materials and methods section.

9)     The PMV model requires for the calculation the parameter of clothing insulation in m2 K/W units. However, we would ask the authors to consider adding in the text the equivalent of the used clothing insulation (0.155 m2 K/W) in the clo units as well. The reason for that is that the clothing insulation is presented with the clo units (or both m2 K/W and clo units) in most of the tables of ASHRAE and ISO standards.

10)  Since there is not much information about the experimental procedure, we assume that the results are derived from one day experiment. However, the actual human thermal perception (actual sensation vote) involves or is influenced by psychological factors (in addition to the other parameters that define the thermal environment; temperatures, humidity etc). How do you know that the results would be the same, if the same experiment was performed the next day or a couple of days later after the initial experiment? In lines 297-298 you are referring to the small group of workers as a limitation. The one-day experiment is also a limitation and should be mentioned clearly and upfront, and in the discussion.

11)  Why do not you combine/correlate each of the measured parameters (air temperature, humidity, air velocity, radiant temperature or even PMV/PPD) with the corresponding question/questions presented in Table 2? For example, the air temperature with the first three questions or with just one of them; or the PMV/PPD with the Q17 question.

Round 2

Reviewer 2 Report

The authors have addressed all my comments. 

Reviewer 3 Report

All suggestions/comments have been addressed appropriately.